# Is Agricultural Emissions Mitigation on the Menu for Tea Drinkers?

**Rebecca Boehm** [1,*]**, Hannah Kitchel** [2]**, Selena Ahmed** [3]**, Anaya Hall** [4]**, Colin M. Orians** [5]**,
John Richard Stepp** [6]**, Al Robbat, Jr.** [7]**, Timothy S. Griffin** [2] **and Sean B. Cash** [2]

1    Food and Environment Program, Union of Concerned Scientists, 1825 K Street NW, Suite 800, Washington, DC 20006, USA
2    Friedman School of Nutrition Science and Policy, Tufts University, 150 Harrison Avenue, Boston, MA 02111, USA
3    College of Education, Health and Human Development, Montana State University, Bozeman, MT 59717, USA
4    Energy and Resources Group, University of California at Berkeley, 310 Barrows Hall, Berkeley, CA 94720-3050, USA
5    Department of Biology, Tufts University, Robinson Hall, Room 364, 200 College Avenue, Medford, MA 02155, USA
6    Department of Anthropology, 1112 Turlington Hall, University of Florida, Gainesville, FL 32611-7350, USA
7    Department of Chemistry, Sensory and Science Center, Tufts University, 200 Boston Avenue, Medford, MA 02155, USA
*    Correspondence: rboehm@ucsusa.org

**Abstract:** Consumers are increasingly concerned about the environmental and social impacts of their purchases. Prior research has assessed willingness to pay (WTP) for environmental and ethical attributes on foods and beverages such as locally grown, fairly traded, and organically produced. However, few studies have examined WTP for agricultural greenhouse gas (GHG) mitigation, especially in the U.S. and to date, no prior study has examined how knowledge or concerns about climate change motivate WTP for climate-friendly products. The objective of this study was to estimate WTP for agricultural GHG mitigation and examine variability in WTP across consumer characteristics, climate change knowledge and risk perception. A sensory-grounded choice experiment and survey assessing climate change knowledge and risk perception was administrated to specialty food and beverage shoppers in the Midwest and Northeastern U.S. Male and lower-income participants, as well as those at the Midwestern study site were willing to pay a higher premium for agricultural GHG mitigation, relative to females, higher income participants, and those in the Northeastern U.S. Knowledge of climate change and level of concerns for the risks it poses were not significantly associated with increased WTP for agricultural GHG mitigation. This suggests that if consumer demand is going to play a role in driving agricultural GHG mitigation, motivations for such purchasing behavior must be more fully understood.

**Keywords:** climate change; willingness to pay; climate change knowledge and risk perception; greenhouse gas emission labels; choice experiment; latent class analysis; carbon footprint

## 1. Introduction

Consumers are increasingly concerned about the environmental and social impacts of their purchases, with companies quickly responding to such demands [1]. Over the past decade there has been an explosion in consumer-facing product labels, especially those promoting sustainable and ethical credence attributes [2,3]. In 2010, approximately 7000 products in the U.S. exhibited an environmental claim, including 89 products asserting to be carbon neutral [4]. These labels are increasingly found on

foods and beverages and can include labels denoting whether the product was organically or locally grown, genetically modified organism-free, fairly traded or designate its carbon footprint. While there is an active and large body of research assessing consumer demand and motivations for all of these types of labels, the area focused on carbon footprint and other climate-related labels is relatively small, especially in the U.S., but it is steadily increasing. With climate change posing a major threat to global social, economic and natural systems, including the food system, all emission mitigation strategies must be identified and exploited. This includes ways to shift consumer demand towards climate-friendly foods and beverages, which can reduce emissions in the agricultural and food system in the short term [5,6]. Consequently, is it imperative to continue to investigate consumer interest and motivations for purchasing low-carbon foods and beverages in order to grow the market for such products.

Despite this growing awareness, consumer interest and motivation for climate-friendly foods and beverages has been studied less extensively than consumer interest in other sustainable and ethical product attributes. In some global markets, food and beverage packages now have labels denoting the level of carbon dioxide emissions generated across the product's life cycle [2,7]. In the markets where these labels exist, consumers have been reported to pay premiums for them [8–15]. For example, Tait et al. (2011) found that United Kingdom and Japanese consumers were willing to pay 1% more for fruit with a 21%–39% reduction in carbon emissions. More recently, German consumers were willing to pay over 30% more for carbon-neutral milk and apple juice [9]. Consumers in South Korea were also willing to pay premiums of 30% or higher for apples with carbon-emission reductions [13]. A study of European consumers found 7%–20% higher willingness to pay than the average market price for milk with reduced carbon emissions [14]. Among Australian consumers, purchases of grocery items with higher than average carbon footprints were significantly lower when carbon footprint labels were added to store shelves [16].

To date, few studies have specifically focused on U.S. consumer willingness to pay for foods or beverages with a reduced-carbon footprint or for agricultural practices that avert the production of climate-warming gases. One U.S.-based study found that consumers were not willing to pay more for apples labeled as locally grown with reduced greenhouse gas emissions from transportation compared to non-local apples [10]. U.S. consumers reported paying a premium for reducing their total carbon footprint, but many consumers also reported not having enough knowledge about carbon footprint measures to make informed, product-specific decisions [17]. A more recent study found that U.S. consumers were willing to pay a price premium for beef produced with prescribed grazing since compared to conventional beef production, it can generate in some instances fewer greenhouse gases [18]. Continued research on U.S. consumer interest in carbon labels and low-carbon foods and beverages is particularly important since increasingly, foods, beverages, and other products will carry labels denoting their carbon footprint. With companies such as Walmart, Nestlé, and Coca-Cola responding with large GHG reduction commitments, such consumer trends are being recognized by the industry, and carbon labels may soon be common in food and beverage markets [19–21].

It is equally important to understand not just if consumers will pay more for low-carbon foods and beverages but what specifically motivates them to make such purchases. Studies have found that willingness to pay a premium for environmental or ethical attributes is heterogeneous across consumer characteristics, including income, self-reported health, education, age, and food shopping preferences [22–27]. Other research shows that shopping habits, health concerns, and knowledge of environmental issues including climate change are associated with willingness to pay premiums [2,28–31]. However, studies do not consistently find that positive attitudes toward improving society or the environment are correlated with consumer willingness to pay more for such product attributes, and the literature has pointed out a consistent gap between consumer purchase intentions and what they actually purchase [2,32–35]. In the U.S. context, little is known about the link between U.S. consumer motivations for purchasing carbon-friendly food and beverage products.

An important factor that may affect consumer willingness to pay more for products with environmentally, climate-, or ethically oriented credence attributes is risk discounting. Environmental issues such as climate change are often viewed as having uncertain consequences, a delayed onset, and are perceived to be less probable in an individual's locale [36]. This discounting can present a psychological barrier to environmentally sustainable behavior, as greater discounting of environmental risk could lead to less environmentally sustainable viewpoints and behaviors [37]. It is thus of interest to understand how people evaluate the risks that climate change poses to them or others, and whether this influences their willingness to pay more for climate-friendly products. Consumers' purchasing behavior could stem from their assessment of the reality of their personal or the global climate change threat, and, therefore, understanding such behavior could prove insightful on how to encourage more environmentally sustainable and climate-friendly purchases. This may be especially true for food and beverage purchases since agricultural production both contributes to and is harmed by a changing climate.

The objective of the present study was to determine if U.S. specialty food and beverage shoppers are willing to pay a price premium for mitigation of agricultural greenhouse gas emissions (GHGs) using green tea as a case study product. The secondary objective was to determine how consumer demographic characteristics, consumer climate change knowledge and risk perception influence willingness to pay for green teas that have been produced with lower levels of GHGs at the agricultural production stage of the supply chain.

## 2. Materials and Methods

### 2.1. Green Tea as a Case Study Product

Green tea was used as a case study product for four specific reasons. First, tea is the mostly widely consumed beverage in the world after water [38] and rates of consumption in the United States have been increasing over the last two to three decades, especially green tea [39,40]. Second, specialty shoppers are an important segment of U.S. demand and these shoppers are seeking out products that they can engage with: they want to know the story behind the product, including its history, country of origin, and the management and manufacturing techniques used to produce it [40]. Tea consumers also interested in the healthfulness and environmental sustainability of tea, so understanding tea's carbon footprint may be an attribute of interest to them, especially as concerns about climate change increase [40]. Third, tea, including green tea, is grown on approximately 4.1 million hectares of land globally, which is almost the same area used to grow the world's supply of fresh fruit [41]. So while tea is a relatively low carbon product compared to coffee and other foods [42], it's aggregate emissions contribution is not inconsequential. Finally, the tea industry is centered in lesser developed countries and serves as an important economic development engine for poorer, rural regions [43,44]. Identifying product characteristics in tea that higher income consumers will pay a premium for can help improve the livelihoods of lower-income farmers and the regions in which they live.

### 2.2. Study Sites and Participant Recruitment

Participants were recruited into the study at three sites in the Boston, Massachusetts metropolitan area between April and August 2015 and at one site in Bozeman, Montana in September 2015. Sites were selected because they sold a variety of tea products, including boxed and bulk green tea and other teas, tea kettles, and other tea accessories at varying price and quality levels. They were also selected because they marketed themselves as specialty and natural food retailers, each offering a variety of products with sustainable and ethical product attributes, including organic certification, fair trade certification, as well as locally and regional grown foods and beverages. These sites include: (1) a small natural foods and supplement retailer in the Boston area, (2) a cooperative grocery store with two locations in the Boston area, (3) a weekly farmers' market in an inner suburb of Boston, and (4) a tea house in Bozeman, MT.

The study was advertised to site customers through electronic newsletters sent out in advance of data collection. Site customers could sign-up to participate in the study in advance for pre-set time slots through an online form or by contacting the research team by phone. Pre-set time slots were arranged at various times of day and days of the week (including weekends) to ensure data was collected from a diverse sample of customers. Advertisements described the length of the study (15 to 20 min) and the compensation for participating (i.e., gift cards to the store where the participant completed the choice experiment). Shoppers were also recruited into the study on a rolling basis at each site during on-site data collection.

The protocol was reviewed and approved as human subjects research by Tufts University and Montana State University Institutional Review Boards.

### 2.3. Choice Experiment Overview

A choice experiment was used to elicit consumer preference and willingness to pay (WTP) for brewed green tea attributes. A choice experiment is a stated preference method to quantitatively estimate consumer value for product characteristics [45]. In choice experiments, participants choose between two or more product alternatives that have different combinations of attributes and attribute-levels. For example, an attribute presented in a choice experiment could be whether or not the product was organically grown, and the attribute-levels would be 'organically grown' or 'not organically grown', thus the attribute would have two levels. Participants choose their most preferred product alternatives in multiple choice sets presented to them in the experiment. Data generated from their choices across multiple choice sets are used to estimate marginal WTP for product attributes. Attributes and levels presented in choice experiments can be real (i.e., available for purchase in a store or market) or hypothetical. Estimating WTP for hypothetical attributes can help predict if consumers would demand such an attribute if it was to become available in the market.

### 2.4. Process Used to Select Choice Experiment Attributes and Attribute-Levels

Feedback collected from qualitative, semi-structured focus groups and assessment of tea attributes and levels available for sale at the study sites was used to determine which non-hypothetical attributes and levels would be most relevant and salient to participants in the choice experiment. Qualitative semi-structured focus groups (six groups, 40 participants total) were conducted over a 2-month period prior to the in-store choice experiment. Focus groups were composed of customers shopping at the four study sites. Participants in the focus groups were asked questions about the types of attributes they preferred on green and other teas they purchase. Overall, focus group participants expressed interest in a variety of product attributes related to health, environmental sustainability, and ethical sourcing. Focus group participants also expressed specific interest in the nutritional quality of the tea (as measured by the presence of antioxidants in tea), the scale of tea production (e.g., small or large farm grown tea), and the season of harvest.

Focus groups were also used to determine if an attribute denoting agricultural mitigation of GHGs would be understandable to and of interest to study participants, since this attribute is not currently available for sale at the study sites or in U.S. tea markets. Focus group participants were asked if the amount of GHGs that were produced in tea production or by the production of other foods mattered to their tea or other food purchasing decisions. They were also asked if they would be interested in a labelling scheme denoting a food or beverage producer's efforts to mitigate emissions. Focus group participants expressed interest in such attributes and noted general concern about climate change and agriculture's contribution to the production of climate-warming gases.

An assessment of the attributes on tea products available at the study sites was also conducted to determine the types of attributes and levels to include in the choice experiment. Based on this informal assessment, the most common attributes available on tea, included: tea type (i.e., green, black, etc.), tea brand, organic certification label, fair-trade certification labels, nutrition fact panels and ingredient

information, and price. In this assessment, it was also determined that tea was most commonly sold in boxes of pre-made tea sachets.

The set of attributes and levels used in the final choice experiment were determined based on information collected in focus groups and in-store assessments. Table 1 provides the attributes and levels presented to participants in the final choice experiment. Figure 1 illustrates a choice set participants completed in the choice experiment. Each participant completed eight choice sets and the presentation of attributes and levels were fully randomized across choice sets and participants. The choice set employed a full-factorial design using the Conjoint Survey Design Tool [46].

**Table 1.** Choice experiment tea product attributes and attribute-levels.

| Attribute | Attribute-Levels |
|---|---|
| Price for 18 tea sachet box | Range: $1.99 to $12.99 in $1 increments |
| Agricultural GHG mitigation | Level 0: Farmer lowers carbon footprint of tea production by reducing use of nitrogen fertilizers that cause greenhouse gases.<br>Level 1: Farmers make no attempt to lower carbon footprint of tea production. |
| Scale of tea production | Level 0: Small-scale production<br>Level 1: Large-scale production |
| Nutritional quality | Level 0: Low antioxidant content<br>Level 1: High antioxidant content |
| Fair trade certification | Level 0: Not fair trade certified<br>Level 1: Fair trade certified |
| Organic certification | Level 0: Not certified organic<br>Level 1: Certified organic |
| Season harvested | Level 0: Pre-monsoon season<br>Level 1: Monsoon season |

Based on this information and what is provided below, which of these choices would you prefer to buy if given the choice?

| | Tea 1 | Tea 2 |
|---|---|---|
| **Nutritional quality** | High antioxidant content | High antioxidant content |
| **Organic certified** | Certified organic | Not certified organic |
| **Carbon footprint** | Farmer lowers carbon footprint of tea production by reducing use of nitrogen fertilizers that cause greenhouse gases. | Farmer lowers carbon footprint of tea production by reducing use of nitrogen fertilizers that cause greenhouse gases. |
| **Price (for box of 18 bags)** | $3.99 | $10.99 |
| **Fair trade certification** | Not fair trade certified | Fair trade certified |
| **Season harvested** | Pre-monsoon season | Monsoon season |
| **Scale of production** | Small scale | Large scale |

|  Tea 1 | Tea 2 | I prefer neither option |
|---|---|---|
| O | O | O |

**Figure 1.** Illustration of the choice sets faced by consumers in the study.

## 2.5. Grounding the Choice Experiment in a Sensory Evaluation of Tea

The choice experiment was combined with tastings of green tea samples since valuation of products has been shown to be linked to sensory characteristics [47,48]. Prior to beginning the choice experiment, participants tasted two green tea samples that the research team obtained from a tea farm in Yunnan, China. These tea samples were identical except one was harvested in the pre-monsoon

season and the other was harvested during the monsoon season. Consumers were required to taste both the pre-monsoon and monsoon harvested samples because a concurrent study was assessing the sensory characteristics and consumer WTP for tea harvested in different seasons. In addition, feedback from the focus groups and prior research indicates that harvest season is especially important to the quality and flavor of green teas [49–51]. A detailed explanation of the tea tasting protocol can be found in the Supplementary Materials.

After tasting tea samples, participants read an instruction sheet (see Supplementary Materials Figure S1) prior to beginning the choice experiment. The instruction sheet explained that participants should consider a real-world setting where they would be buying a box of one of the sampled teas containing 18 sachets and to choose the product they preferred most in this setting, or they could choose the "I prefer neither option".

### 2.6. Assessment of Climate Change Knowledge and Risk Perception

After completing the choice experiment, participants answered questions in a survey about their knowledge of climate change and the risks that climate change poses to them, society, and to the environment. These questions were derived from O'Connor et al., 1999. To assess each participants' climate change knowledge and assign them a quantitative knowledge score, participants were asked to decide which phenomena is a major cause, minor cause, or not a cause of climate change. These phenomena included: "pollution/emissions from business/industry"; "people driving cars"; "use of coal/oil by utilities/electric companies"; "use of aerosol spray cans"; "chemicals that destroy insects and pests"; "depletion of the ozone layer"; "nuclear power generation"; "people heating/cooling their homes"; "destruction of forests". Up to two points were awarded per question when participants answered correctly. Participants could obtain a knowledge score of zero points if they answered none of the questions correctly, or up to 18 points if they answered all questions correctly.

To assess risk perception separately from knowledge of climate change, participants were asked the following question (also from O'Connor et al., 1999): "Suppose the average global temperature increases by 3–4 degrees Fahrenheit over the next 50 years as a result of climate change, how likely do you think the following events will be?". The six events included: "many people's quality of living will decrease"; "my quality of living will decrease"; "starvation will occur in much of the world"; "starvation will occur where I live"; "rates of serious disease will increase"; "my chance of suffering from a serious disease will increase". A 7-point Likert scale was used to assess their perception of the risk that climate change poses for these events, where 1 indicated "Very Unlikely" and 7 indicated "Very Likely". Higher scores chosen on the Likert scale indicated that a participant was more concerned about the risk of climate change than if they chose a lower score on the Likert scale. From these questions, a single, continuous measure of risk perception was generated using principal component analysis (PCA), which is used to convert multiple Likert scale responses to one variable that measures different key components of individual attitudes and beliefs [52].

### 2.7. Demographic and Tea Purchase and Consumption Questions

Participants reported their demographic characteristics including: annual income (in U.S. dollars), gender, race, ethnicity, age, educational attainment and location where they completed the survey (either Boston, MA metropolitan area or Bozeman, MT). Based on self-reported income, participants were classified as high or low income using the U.S. Census Bureau American Community Five Year Estimates from 2011 to 2015 median income values for the Boston metropolitan area and Bozeman, MT [53]. Participants whose incomes were above the median income values were classified as high income.

The choice experiment and survey questions were administrated to participants using Qualtrics survey software.

*2.8. Data Analysis*

Data analysis was conducted in Stata 15.1/SE.

**Estimating Willingness to Pay (WTP) for Tea Product Attributes**

Estimating WTP from choice experiment data relies on the assumptions of Lancastrian consumer theory and Random Utility Theory (RUT) [54,55]. These theories assume that consumers seek a product yielding the highest utility, and the utility they derive from choosing the product with the highest utility can be decomposed as a linear combination of a product's attribute-levels and a random error component. Consequently, the probability of selecting a product alternative in a choice set can be modeled as a linear function of the attributes and levels presented to the participant in a choice experiment, where:

$$U_{ij} = \hat{U}_{ij} + \varepsilon_{ij} \tag{1}$$

$\hat{U}_{ij}$ is the utility derived from participant $i$'s product selection in a choice set $j$ and $\varepsilon_{ij}$ is the random error component. $\hat{U}_{ij}$ is a linear function of the attributes for each product in the choice set,

$$\hat{U}_{ij} = \mathbf{X}_{ij}\boldsymbol{\beta} \tag{2}$$

where $\boldsymbol{\beta}$ represents a vector of parameters to be estimated and $\mathbf{X}$ represents a vector of variables representing the attributes and their levels available in choice set $j$.

Theoretically, the probability of a product being chosen from choice set $j$ is equal to the probability that the utility of that product is greater than all other products in the choice set. This can be described more precisely as:

$$P_{ij} = Prob\big(U_{ij} \geq U_{ik}\big), \ for \ all \ k \ \in \ C_i \ with \ k \ \neq \ j) \tag{3}$$

where $U_{ij}$ is the utility derived from the product chosen in choice set $j$.

Using RUT as the analytical foundation, a logistic regression model is used to estimate the probability of consumer $i$ choosing alternative $k$ in choice set $j$. The probability of selection can be formally expressed as:

$$Prob_i \ \{y_i = j\} = \frac{e^{x_i\beta}}{\sum_{j=1}^{j} e^{x_i\beta}} \ with \ j \ \in \ C_i \tag{4}$$

The linear function is then estimated with maximum-likelihood simulation, where:

$$\begin{aligned}
U_{ij} \\
= \ &\beta_0 + \beta_1 Price_{ij} + \beta_2 Ag. \ GHG \ mitigation_{ij} + \ \beta_3 Antioxidants_{ij} \\
&+\beta_4 Scale \ of \ Production_{ij} + \ \beta_5 Harvest \ Season_{ij} \ + \ \beta_6 Fair \ trade \ certified_{ij} \\
&+\beta_7 Organic \ certified_{ij} + \ \beta_8 Ag. \ GHG \ mitigation_{ij} \\
&*knowledge_i \ + \ \beta_9 Ag. \ GHG \ mitigation_{ij} * risk_i + \beta_{10} Neither + \ \beta_{11} Neither_{ij} \\
&*knowledge_i \ + \ \beta_{12} Neither_{ij} * risk_i \ + \boldsymbol{\beta_s} \sum \ sociodemographics_i + \ *Neither_{ij} + \ \varepsilon_{ij}
\end{aligned} \tag{5}$$

*Agricultural GHGs mitigation, Antioxidants, Scale of Production, Season of Harvest, Fair Trade Certified, Organic Certified* were binary variables and their corresponding levels are described in Table 1. The price variable was a continuous measure, as described in Table 1. *Knowledge$_i$* was a variable measuring participant climate change knowledge and *risk$_i$* is the variable measuring participant risk perception of climate change, the latter of which was computed as a single measure using principal component analysis to ensure within-participant consistency of responses [52]. The *Neither* variable was binary and constructed like an alternative specific constant variable to describe the labels used to identify tea alternatives in each choice set (i.e., "Tea 1", "Tea 2", and "I prefer neither option"). Neither was equal to one if the participant selected "I prefer neither option" and the variable was zero otherwise. *β$_s$* represents a vector of coefficients for each consumer sociodemographic characteristics. WTP for

each attribute-level combination was calculated as the ratio of each attribute's coefficient estimated in the conditional logistic regression model divided by the estimated price attribute coefficient.

**Identifying preference heterogeneity for attributes and association with participant demographic characteristics, climate change knowledge and risk perception scores**

Latent class analysis was used to assess preference heterogeneity for product attributes and to determine whether heterogeneity in product selections in the choice experiment was associated with participant characteristics, climate change knowledge and risk perception scores. As opposed to mixed logit models, which assume preference heterogeneity takes a continuous distribution, latent class models assume that consumers can be segmented into discrete categories or groups based on preference heterogeneity. These consumer segments can then be considered in the context of real market settings in a more intuitive fashion than results from mixed logit models.

Latent class analysis identifies unobservable–or latent–subgroups of participants in a sample, based on choice experiment responses [56,57]. In latent class analysis, participants are sorted into $s$ classes and participants are assumed to have homogenous preferences within classes but heterogeneous preferences across classes [58]. To determine latent class membership, the probability that participant $i$ chooses alternative $j$ is estimated as in Equation (4), but modified to account for latent class-specific utility parameters:

$$P\left(x_{ijk}\middle|s\right) = \frac{e^{x_i\beta}}{\sum_{k=1}^{j} e^{x_i\beta}} \tag{6}$$

Then, for a given class assignment $s$, the probability of participant $i$ making a series of choices would have the joint probability as:

$$P_i(s) = \prod_{j=1}^{J}\prod_{k=1}^{K}\left(\frac{e^{x_i\beta}}{\sum_{k=1}^{j} e^{x_i\beta}}\right)^{y_{ijk}} \tag{7}$$

where $y_{ijk}$ is equal to 1 if participant $i$ chooses alternative $j$ in choice set $k$, and 0 otherwise.

A fractional multinomial logit model is estimated to determine the probability that participant $i$ falls into class $s$, using the following equation:

$$\lambda_{is}(\theta) = \frac{e^{\theta_s z_i}}{1 + \sum_{k=1}^{S-1} e^{\theta_s z_i}} \tag{8}$$

where $z$ are observed choices made by participant $i$ and $\theta$ is the vector of class membership parameters to be estimated [56].

Then, the exact probability that participant $i$ belongs to class $s$ is estimated:

$$ln(S) = \sum_{i=1}^{I} ln \sum_{s=1}^{S} \lambda_{is}(\theta)P_i(s) \tag{9}$$

The log likelihood function is then estimated for participants in the sample by summing their log likelihoods:

$$M = \sum_{i=1}^{I} ln \sum_{s=1}^{S} \lambda_{is}(\theta)P_i(s) \tag{10}$$

Finally, posterior estimates of the probability that participant $i$ belongs to class $s$ evaluated at the $s$th iteration is calculated using Bayes theorem [59].

$$T_{is}(\theta^s) = \frac{P_i(s)\lambda_{is}(\theta^s)}{\sum_{s=1}^{S} P_i(s)\lambda_{is}(\theta^s)} \tag{11}$$

Aikaike Information Criteria (AIC), Consistent Aikaike Information Criterion (CAIC), and Bayesian Information Criterion (BIC) were used to determine the optimal number of latent classes to segment participants [56].

Multiple linear regression was then used to determine how probability of latent class membership was associated with participant demographic characteristics, climate change knowledge and risk perception. The dependent variable $P(membership)$ was the posterior probability that participant $i$ was estimated as a member of latent class $s$ ($s = 1, 2, \ldots, S$). Independent variables in these models included participant demographic characteristics, knowledge and risk perception scores.

## 3. Results

### 3.1. Demographic Characteristics of Participants

A total of 380 participants were included in the final analysis sample. Table 2 presents demographic characteristics of study participants. The sample was predominately female (74.2%), Caucasian/White (81.8%), and people aged 34 years or younger (75.3%). A total of 68.0% of the sample had earned a college degree or higher and 24.0% of the sample was classified as high income. The mean knowledge score was 11.6 (SE 0.022) and the mean score for our risk perception measure was 4.5 (SE 0.011).

**Table 2.** Participant demographic characteristics (n = 380).

|  | % of Total |
|---|---|
| **Gender** | |
| Female | 74.2 |
| Male | 25.8 |
| **Age Group** | |
| 18–24 | 31.3 |
| 25–34 | 44.0 |
| 35–44 | 11.6 |
| 45–54 | 8.0 |
| 55–64 | 3.7 |
| > 65 years | 1.6 |
| **Education Level** | |
| No college | 32.1 |
| Some college completed, but no degree earned | 54.0 |
| College graduate or higher | 14.0 |
| **% High Income** | 24.5 |
| **Racial or ethnic group** | |
| White | 81.8 |
| Asian | 4.2 |
| Black or African American | 8.2 |
| Other race/multiple race | 5.8 |
| Hispanic or Latino | 4.3 |
| **Location of survey participant** | |
| Boston, MA | 79.0 |
| Bozeman, MT | 21.0 |

### 3.2. Association between Probability of Selecting Tea Alternative and WTP for Agricultural GHG Mitigation

Results from the conditional logistic regression models are reported in Table 3. In Model 1, including only attributes associated with tea alternatives presented in the choice set, all coefficients were statistically significant at the 1% level, including the coefficient for the "I prefer neither" tea option. The sign of the coefficient for the "I prefer neither tea option" was also negative and statistically

significant, indicating that, on average, participants in the sample had a preference for the tea options available in the choice sets presented. The coefficient for price was also negative, as expected, since the law of consumer demand dictates that as price increases, consumer preference for a product, all else being equal, will decrease. The coefficient for agricultural GHGs mitigation was positive, suggesting that participants, on average, preferred teas with reduced GHGs in tea production compared to no mitigation. The magnitude of the coefficient for agricultural GHGs mitigation was larger than the coefficients for all other attributes presented in the choice experiment, indicating that the presence of the agricultural GHGs mitigation claim more strongly influenced the probability that a participant selected a tea compared to other attributes presented in the choice experiment.

### 3.3. Influence of Participant Demographic Characteristics, Climate Change Knowledge and Risk Perception on Tea Alternative Selection

Table 3 includes results from Model 2, the conditional logistic regression that includes participant demographic characteristics as covariates interacted with the agricultural GHG mitigation attribute variable and results from Model 3 which includes knowledge and risk perception score variables interacted with the agricultural GHGs mitigation attribute variable.

Model 2 results indicate that high-income participants had less interest in reduced GHG in tea production compared to low-income participants, since the sign on the interaction term for reduced GHGs in tea production and high income is negative and statistically significant. Participants recruited at the Bozeman, MT site had more interest in reduced GHG in tea production compared to participants completing the survey in the Boston area. Male participants also had stronger interest in reduced GHGs in tea production compared to females.

Model 3 results indicate that, on average, the probability of selecting a tea alternative was significantly associated only with a participant's risk perception score. However, the coefficients for reduced GHGs in tea production interacted with the knowledge and risk perception scores were not statistically different from zero, indicating that climate change knowledge and risk perception scores were not significantly associated with participant interest in reduced GHGs in tea production.

**Table 3.** Conditional logistic regression model estimated coefficients (standard errors in parentheses) showing association between probability of selecting a tea alternative and tea attribute-levels, participant demographic characteristics, and climate change knowledge and risk perception scores.

| | Model 1: Tea Attributes only | | Model 2: Tea Attributes + Participant Demographic Characteristics | | Model 3: Tea Attributes + Participant Demographic Characteristics + Knowledge and Risk Score Interactions | |
|---|---|---|---|---|---|---|
| | Coefficients Reported (Standard Errors in parentheses) | | | | | |
| **Product attributes** | | | | | | |
| Price (continuous measure) | −0.106 ** | (0.00780) | −0.108 *** | (0.00786) | −0.106 *** | (0.00780) |
| Agricultural GHGs mitigation (no mitigation is referent) | 0.645 ** | (0.0563) | 0.955 ** | (0.378) | 0.698 *** | (0.226) |
| Large scale production (small-scale production is ref.) | −0.229 ** | (0.0560) | −0.235 *** | (0.0563) | −0.228 *** | (0.0560) |
| High antioxidants (low antioxidant is ref.) | 0.363 ** | (0.0561) | 0.359 *** | (0.0564) | 0.362 *** | (0.0561) |
| Fair trade certified (not fair traded is ref.) | 0.474 ** | (0.0561) | 0.473 *** | (0.0564) | 0.473 *** | (0.0562) |
| Organic certified (not organic certified is ref.) | 0.490 ** | (0.0561) | 0.493 *** | (0.0564) | 0.489 *** | (0.0561) |
| Pre-monsoon tea (monsoon tea is ref.) | 0.574 ** | (0.0562) | 0.574 *** | (0.0565) | 0.575 *** | (0.0562) |
| I prefer neither tea option (tea 1 and tea 2 are ref.) | −0.605** | (0.0996) | −0.216 | (0.391) | −0.350 | (0.255) |
| **Probability of tea selection by knowledge and risk perception scores** | | | | | | |
| Knowledge score (continuous measure) | | | | | −0.00335 | (0.0226) |
| Risk score (continuous measure) | | | | | −0.0546 ** | (0.0237) |
| **Probability of tea selection by demographic characteristics** | | | | | | |
| **Male (female is ref.)** | | | −0.364 *** | (0.105) | −0.395 *** | (0.107) |
| **Age** | | | | | | |
| 25–34 (<25 years is ref.) | | | −0.949 *** | (0.319) | −1.019 *** | (0.326) |
| 35–44 | | | −1.010 *** | (0.315) | −1.079 *** | (0.323) |
| 45–54 | | | −0.191 | (0.323) | −0.256 | (0.328) |
| 55–64 | | | −0.379 | (0.337) | −0.473 | (0.343) |
| >64 | | | −0.359 | (0.364) | −0.433 | (0.370) |
| **Race or ethnic group** | | | | | | |
| Black (white is ref.) | | | −0.441 *** | (0.170) | −0.433 ** | (0.171) |
| Asian | | | −0.768 *** | (0.271) | −0.814 *** | (0.274) |
| Other race | | | −0.608 *** | (0.223) | −0.598 *** | (0.224) |
| Hispanic/Latino (non-Hispanic is ref.) | | | 0.119 | (0.274) | 0.120 | (0.275) |
| **High Income (Low income is ref.)** | | | 0.322 *** | (0.0999) | 0.317 *** | (0.100) |
| **Boston, MA (Bozeman, MT location is ref.)** | | | 0.843 *** | (0.135) | 0.887 *** | (0.138) |
| **High level of education attained** | | | | | | |

**Table 3.** *Cont.*

| | Model 1: Tea Attributes only | Model 2: Tea Attributes + Participant Demographic Characteristics | | Model 3: Tea Attributes + Participant Demographic Characteristics + Knowledge and Risk Score Interactions | |
|---|---|---|---|---|---|
| | | **Coefficients Reported (Standard Errors in parentheses)** | | | |
| Some college completed, but no degree earned (No college is ref.) | | 0.0489 | (0.141) | 0.0365 | (0.141) |
| College degree or higher (No college is ref.) | | 0.175 | (0.135) | 0.143 | (0.135) |
| **Preference for agriculture GHGs mitigation by knowledge and risk perception scores** | | | | | |
| Agricultural GHGs mitigation * Knowledge score | | | | −0.00454 | (0.0188) |
| Agricultural GHGs mitigation * Risk score | | | | 0.0131 | (0.0204) |
| **Preference for agriculture GHGs mitigation by demographic characteristics** | | | | | |
| **Male (female is ref.)** | | 0.202 ** | (0.0901) | | |
| **Age** | | | | | |
| 25–34 (<25 years is ref.) | | 0.0544 | (0.321) | | |
| 35–44 | | −0.0351 | (0.317) | | |
| 45–54 | | −0.429 | (0.329) | | |
| 55–64 | | −0.129 | (0.343) | | |
| >64 | | −0.512 | (0.371) | | |
| **Race or ethnic group** | | | | | |
| Black (white is ref.) | | 0.0490 | (0.179) | | |
| Asian | | 0.263 | (0.264) | | |
| Other race | | 0.177 | (0.222) | | |
| Hispanic/Latino (non-Hispanic/Latino is ref.) | | 0.102 | (0.241) | | |
| **High Income (Low income is ref.)** | | −0.236 ** | (0.0976) | | |
| **Boston, MA (Bozeman, MT location is ref.)** | | −0.289 *** | (0.105) | | |
| **Highest level of education attained** | | | | | |
| Some college (No college is ref.) | | 0.0173 | (0.129) | | |
| College degree or higher (No college is ref.) | | −0.135 | (0.119) | | |
| Observations (380 participants, each with 24 total choice options in the experiment) | 9120 | 9120 | | 9120 | |
| $\chi^2$ | 722.6 | 934.8 | | 901.1 | |
| P($\chi^2$) | <0.0001 | <0.0001 | | <0.0001 | |
| Log-likelihood | −5414 | −5308 | | −5325 | |

**Notes:** Standard errors in parentheses, coefficient values statistically significant from zero denoted by: *** $p < 0.01$, ** $p < 0.05$.

*3.4. Average Willingness to Pay for Agricultural GHGs Mitigation Relative to Other Attributes*

Average WTP and 95% confidence intervals for each attribute included in the base model (Model 1, Table 3) are summarized in Table 4. Participants were willing to pay the highest premium for Agricultural GHG mitigation compared to all other attributes presented in the choice experiment. Additionally, average WTP for agricultural GHGs mitigation was not statistically different than average WTP for fair trade and certified organic tea, and pre-monsoon harvested tea. Average WTP for agricultural GHGs mitigation was statistically higher than WTP for tea with high antioxidant levels and small-scale tea production.

**Table 4.** Average willingness to pay for agricultural GHGs mitigation and other attributes presented in the sensory-grounded choice experiment (n = 9120; 380 participants, each with 24 total choice options in the experiment).

|  | Average WTP | 95% Confidence Interval |
|---|---|---|
| Pre-monsoon harvested tea | 5.43 | (4.16, 6.70) |
| Small scale production | 2.16 | (1.08, 3.25) |
| Agricultural GHGs mitigation | 6.09 | (4.76, 7.43) |
| High antioxidants | 3.43 | (2.29, 4.56) |
| Fair trade certified | 4.47 | (3.27, 5.69) |
| Organic certified | 4.63 | (3.41, 5.85) |

*3.5. Preference Heterogeneity across Demographic Characteristics, Climate Change Knowledge, and Risk Perception Using Latent Class Analysis*

Latent class analysis results showing heterogeneity in participant preference for the agricultural GHG mitigation and other attributes are presented in Table 5. The optimal number of latent classes was determined to be six, based on the AIC, CAIC, BIC and Log-likelihood values for latent class models with two to six classes (see Table S1 for AIC, CAIC, BIC, and log-likelihood values for classes two through six). Table 5 shows attribute coefficient values for each of the six latent classes and estimated percent of study participants assigned to each class. The two largest classes were class 1 (28.4%) and class 4 (29.5%). For both of these classes, virtually all the attributes in the choice experiment were associated with the probability of selecting a tea alternative. By comparison, classes 2, 5 and 6 did not have such widespread interest in the attributes presented.

There was significant variation in the coefficient values for all attributes presented in the choice experiment across the six latent classes, as presented in Table 5. Coefficient values for agricultural GHGs mitigation varied significantly across classes. The coefficient for agricultural GHGs mitigation was largest for class one and five; these two classes differed in that class one exhibited a preference for the pre-monsoon tea, muted price sensitivity and a preference for fair trade certification, whereas members of class five exhibited a skepticism of a fair-trade attribute and a preference for large-scale production. Members of classes four and six were also more likely to select teas with the agricultural GHGs mitigation attribute, but differed in their interest in the tasted tea products and the high antioxidant attribute. Members of class three were responsive to fair trade and organic claims, but not to the novel claim of agricultural GHGs mitigation. For members of class two, there was no association between the probability of selecting a tea alternative and agricultural GHGs mitigation or other credence attributes presented; these shoppers' choices appeared to be driven solely by price.

**Table 5.** Latent class analysis model estimated coefficients (standard errors in parentheses) for each choice experiment attribute for each of the six latent class groups. Coefficients describe the association between attributes and their associated levels and the probability of participant selecting a tea alternative in the choice experiment. Percent of participants estimated to belong to each class is reported at the bottom of the table.

| Attribute | Class 1 | Class 2 | Class 3 | Class 4 | Class 5 | Class 6 |
|---|---|---|---|---|---|---|
| | **Coefficients Reported (Standard Errors in Parentheses)** | | | | | |
| "I prefer neither" tea | −1.069 *** | 1.739 | 1.792 ** | 0.620 *** | −13.29 | −3.845 *** |
| | (0.389) | (1.561) | (0.875) | (0.235) | (12.15) | (0.858) |
| Pre-monsoon tea | 0.591 *** | 1.069 | −3.132 *** | 1.252 *** | 17.36 | 0.278 |
| | (0.176) | (0.898) | (0.890) | (0.160) | (12.24) | (0.212) |
| Large-scale production | −0.117 | 1.900 | −1.011 ** | −0.413 *** | 1.227 * | −0.354 * |
| | (0.188) | (1.898) | (0.411) | (0.128) | (0.723) | (0.200) |
| Agricultural GHGs mitigation | 1.528 *** | −1.284 | −0.0398 | 0.775 *** | 2.180 *** | 0.641 *** |
| | (0.266) | (1.076) | (0.376) | (0.132) | (0.756) | (0.193) |
| High antioxidants | 0.570 *** | −6.115 | 0.731 * | 0.595 *** | 2.157 ** | 0.353 |
| | (0.166) | (8.128) | (0.438) | (0.130) | (0.906) | (0.239) |
| Fair trade certification | 0.769 *** | −0.197 | 0.637 * | 0.784 *** | −1.911 * | 0.410 ** |
| | (0.209) | (0.928) | (0.358) | (0.133) | (1.061) | (0.209) |
| Organic certification | 0.763 *** | −0.351 | 2.016 *** | 0.668 *** | 2.309 *** | 0.538 *** |
| | (0.167) | (0.910) | (0.472) | (0.137) | (0.772) | (0.208) |
| Price | −0.0449 | −0.350 * | −0.134 *** | −0.126 *** | −0.387 *** | −0.287 *** |
| | (0.0330) | (0.203) | (0.0476) | (0.0178) | (0.141) | (0.0583) |
| Estimated % of participants belonging to each class | 28.4% | 6.8% | 6.1% | 29.5% | 5.1% | 24.1% |
| Observations | 380 | | | | | |

Notes: Standard errors of estimated coefficients reported in parentheses. Coefficients statistically different from zero denoted with: *** $p < 0.01$, ** $p < 0.05$, * $p < 0.10$.

Table 6 shows the association between participant demographic characteristics and the probability of membership to each of the six latent classes. Participant age, gender, and geographic location significantly predicted the probability of membership to multiple latent classes. The probability of being in class 1, the class which had significant interest in almost all attributes presented in the choice experiment, was lower if the participant completed the survey in the Boston, Massachusetts area and if they were aged 35 to 44 years old compared to younger participants. Interestingly, higher income participants had a higher probability of being in class 2, whose choices were driven only by the price attribute. The probability of membership to class 3, characterized by non-responsiveness to the agricultural GHGs mitigation attribute, was higher for females and participants less than 64 years old. Meanwhile, completing the survey in the Boston area and being female predicted membership to class 4, which was the only class responsive to all the attributes presented in the choice experiment. Race and ethnicity and age were predicators of membership to class five, which exhibited the highest responsiveness to the agricultural GHGs mitigation claim compared to other classes. Age only predicted the probability of membership to class 6 which was composed of participants who had interest in agricultural GHG mitigation but not the tea harvest season attribute.

**Table 6.** Multiple linear regression coefficients for each latent class, showing the association between probability of membership to each latent class and participant demographic characteristics.

| Participant Characteristics | Class 1 | Class 2 | Class 3 | Class 4 | Class 5 | Class 6 |
|---|---|---|---|---|---|---|
| | Coefficients Presented (Standard Errors in Parentheses) | | | | | |
| **Age (<25 Years is Referent)** | | | | | | |
| 25–34 years | −0.0132 | −0.0249 | 0.0241 | 0.0455 | −0.0349 | 0.00351 |
| | (0.0413) | (0.0224) | (0.0271) | (0.0484) | (0.0256) | (0.0400) |
| 35–44 years | −0.146 *** | 0.102 * | −0.0227 | 0.136* | −0.00888 | −0.0598 |
| | (0.0484) | (0.0582) | (0.0236) | (0.0749) | (0.0399) | (0.0606) |
| 4–54 years | −0.00739 | 0.0936 | 0.0187 | 0.0285 | −0.0742 *** | −0.0592 |
| | (0.0777) | (0.0635) | (0.0469) | (0.0875) | (0.0231) | (0.0648) |
| 55–64 years | 0.0359 | 0.0123 | 0.0341 | 0.136 | −0.0153 | −0.203 *** |
| | (0.122) | (0.0689) | (0.0669) | (0.142) | (0.0690) | (0.0419) |
| >64 years | 0.0229 | 0.122 | −0.0545 ** | 0.120 | −0.0748 ** | −0.135 |
| | (0.171) | (0.139) | (0.0275) | (0.209) | (0.0302) | (0.109) |
| **Male (female is ref.)** | 0.0566 | −0.0150 | −0.0363 * | −0.104 ** | 0.0351 | 0.0634 |
| | (0.0418) | (0.0240) | (0.0218) | (0.0465) | (0.0301) | (0.0409) |
| **High Income (low income is ref.)** | −0.0389 | 0.0558 * | −0.0193 | 0.0177 | 0.0208 | −0.0360 |
| | (0.0426) | (0.0334) | (0.0231) | (0.0556) | (0.0256) | (0.0405) |
| **Educational attainment (No college is ref.)** | | | | | | |
| Some college completed, but no degree obtained | −0.0359 | 0.00735 | −0.00253 | 0.0597 | 0.000775 | −0.0294 |
| | (0.0402) | (0.0302) | (0.0236) | (0.0487) | (0.0236) | (0.0383) |
| College graduate or higher | −0.00776 | 0.0184 | 0.00775 | 0.0309 | −0.0292 | −0.0200 |
| | (0.0599) | (0.0393) | (0.0317) | (0.0685) | (0.0291) | (0.0552) |
| **Race or ethnic group (White is ref.)** | | | | | | |
| Black | 0.0561 | −0.0229 | 0.0304 | −0.128 | −0.0113 | 0.0762 |
| | (0.0989) | (0.0488) | (0.0400) | (0.111) | (0.0705) | (0.0906) |
| Asian | 0.0542 | 0.00961 | 0.0432 | −0.0510 | −0.0570 *** | 0.00106 |
| | (0.0705) | (0.0414) | (0.0495) | (0.0781) | (0.0160) | (0.0609) |
| Other/multiple race | −0.00618 | 0.0163 | −0.000898 | 0.0856 | −0.00696 | −0.0879 |
| | (0.0754) | (0.0489) | (0.0483) | (0.0995) | (0.0387) | (0.0588) |
| Hispanic or Latino (non-Hispanic/Latino is ref.) | −0.0643 | −0.0280 | −0.0179 | 0.136 | −0.0597 ** | 0.0342 |
| | (0.0885) | (0.0187) | (0.0418) | (0.133) | (0.0250) | (0.104) |
| **Boston, MA site (Bozeman, MT is ref.)** | −0.123 *** | 0.0225 | −0.0812 ** | 0.200 *** | 0.0300 | −0.0486 |
| | (0.0474) | (0.0268) | (0.0355) | (0.0486) | (0.0249) | (0.0453) |
| Observations | 380 | | | | | |

Notes: Statistical significance denoted by * $p < 0.10$ ** $p < 0.05$, *** $p < 0.01$.

Table 7 shows the association between participant knowledge score and the probability of latent class membership to each of the six classes. Results indicate that climate change knowledge and risk perception were not important differentiators of tea selection in the choice experiment. Participant knowledge score and risk perception were negatively associated with probability of membership to class two, but only at the 10% level of significance. Participant knowledge and risk perception scores were not associated with the probability of membership to any other classes.

**Table 7.** Multiple linear regression coefficients and standard errors reported showing the association between probability of membership to each latent class and knowledge and risk perception scores.

| | Class 1 | Class 2 | Class 3 | Class 4 | Class 5 | Class 6 |
|---|---|---|---|---|---|---|
| | Coefficients Reported (Standard Errors in Parentheses) | | | | | |
| Knowledge score | 0.00272 | −0.0107 * | −0.00116 | −0.000545 | 0.00168 | 0.00804 |
| | (0.00852) | (0.00592) | (0.00410) | (0.00937) | (0.00589) | (0.00747) |
| Risk perception score | 0.000195 | −0.0122 * | 0.00103 | 0.00909 | −0.00239 | 0.00431 |
| | (0.00919) | (0.00742) | (0.00427) | (0.0104) | (0.00528) | (0.00890) |
| Observations | 380 | | | | | |

Notes: Standard errors in parenthesis below estimated coefficients; statistical significance denoted by * $p < 0.10$.

## 4. Discussion

This study assessed preference and willingness to pay for agricultural GHG mitigation using green tea as a case study in a choice experiment with a sub-set of U.S. specialty food and beverage shoppers. Of key interest was determining if knowledge of climate change or perceptions of the risks that it pose to individuals or society influences WTP for agricultural GHG mitigation. Mitigating GHGs from all sectors of society is urgently needed to reduce the impacts of rising global temperatures caused by climate change. Consequently, understanding what motivates consumers to buy products—including foods and beverages—that are more climate-friendly is an important step towards decreasing emissions in the U.S. and globally.

Tea was used as a case study product for this analysis, but the methods for the study can assuredly be applied to other foods and beverage products available in a variety of other contexts. In particular, the linkage between consumer concerns about the risks that climate change poses and purchasing behavior is an important area of future research especially, as the impacts of climate change become more apparent.

This study has limitations that warrant discussion. First, the consumer sample was not constructed to represent the U.S. population that consumes tea, but instead was focused on specialty food and beverage shoppers who may be more responsive to sustainability or production-based claims than the average consumer. Thus, WTP values estimated in this study may be different compared to values estimated for a representative sample of U.S. tea drinkers or consumers more generally. Second, willingness to pay estimates using choice experiments can suffer from hypothetical bias, which leads to an over estimation of willingness to pay [60]. However, even if one is skeptical of the magnitude of WTP estimates generated for agricultural GHGs mitigation, the relative ranking of this hypothetical and novel attribute to more familiar ones (i.e., organic certified or fair-trade certified) presented in this choice experiment offers new insights. At the same time, valuation methods that more adequately mitigate hypothetical bias could be employed to more accurately or precisely estimate WTP for agricultural GHGs mitigation. Finally, various measures for assessing climate change knowledge and risk perceptions exist; the measures used in this study were used because they offered a relatively simple and broad assessment of consumer knowledge and concerns about climate change. Future studies could use other more updated or more nuanced measures to assess climate change knowledge and consumer concerns about the risk it poses. A more nuanced or detailed assessment of climate change beliefs and risk perceptions may shed more light on what specifically motivates climate-friendly purchasing behaviors [61]. This is especially important as the impacts of climate change evolve over time.

Results from these analyses indicate that, on average, study participants were willing to pay a premium for agricultural GHGs mitigation, but there was significant variation in preference for this attribute depending on key consumer characteristics. Specifically, males, participants from the Midwestern study site and lower-income individuals had higher WTP for agricultural GHG mitigation. In the latent class analysis, higher household income was significantly associated with the probability of members to latent class two, the class whose members were sensitive to the price levels presented in the choice experiment but no other product attributes. Investigating why higher-income individuals or households had less interest in agricultural GHG mitigation is an important area of future research, especially since higher-income households globally and in the U.S. have higher carbon footprints with respect to food and other consumer activities [62,63]. Another area of further investigation would be in how geographic location influences consumer interest in purchasing more climate friendly foods and beverages, since our results indicate significant differences in willingness to pay for agricultural GHG mitigation across study sites.

Climate change knowledge and risk perception did not significantly influence consumer preference for agricultural GHG mitigation. This finding persisted even when a latent class analysis was used to investigate consumer preference heterogeneity for agricultural GHG mitigation and other attributes presented in the choice experiment. These results raise a question as to why knowledge and risk

perception of climate change did not stimulate participants WTP in agricultural GHG mitigation. These results are consistent with prior studies showing that consumers discount the risk of some environmental issues, such as climate change, since the risks accumulate gradually or are too distal to have an impact directly on their lives [36,64]. However, a 2011 survey of a representative sample of the U.S. population found that motiving mitigation behavior was dependent on the perceived threats of climate change and its severity [65]. The same study found that framing climate change discourse from a health perspective could potentially motivate mitigation behavior even more than messages about threats and risk. Since food and beverage purchasing habits have implications for nutrition, health and the climate, future work could incorporate product attributes more directly related to nutrition and health, or assess in more detail consumer knowledge or risk perceptions about how climate change can impact human health and the quality of foods and beverages.

Another interpretation of these results is that general knowledge of climate change or perceptions of risk are not clearly mapped to food choice. The participants in our study were generally more likely to choose products that included the agricultural GHG mitigation claim, but this was not related to knowledge and risk perception claims, as illustrated in the latent class analysis results. For those seeking to influence food consumers, this may imply that providing product-specific claims may be a more effective means of influencing food choice than general climate education or risk awareness measures.

**Supplementary Materials:** The following are available online at http://www.mdpi.com/2071-1050/11/18/4883/s1, Figure S1: Instruction Sheet for Choice Experiment, Table S1: AIC, CAIC, and BIC information criteria values for latent class models with 2-6 classes.

**Author Contributions:** R.B., A.R.J., C.M.O., S.A., S.B.C., T.S.G., and J.R.S. acquired funding and conceptualized the project; R.B. and S.B.C. designed the experiment; R.B., S.B.C., A.H., H.K. acquired and analyzed the data; R.B. and H.K. performed statistical analysis; R.B., S.B.C., H.K., S.A., T.S.G., J.R.S., C.M.O., A.R.J. contributed to manuscript revision and approved the submitted version.

**Funding:** Primary financial support for the study was provided by the National Science Foundation Coupled Human and Natural Systems program (Award #1313775). Additional funding was provided by the Friedman Family Foundation, Tufts Institute of the Environment, and the National Institute of General Medical Sciences of the National Institutes of Health under Award Number P20GM103474. The content is solely the responsibility of the authors and does not necessarily represent the official views of the National Institutes of Health.

**Acknowledgments:** The authors would like to thank undergraduate and graduate students involved in data collection, cleaning, and processing: Julia Appel, Carla Curle, Iris Levine, Debra Kraner, and Rocio Rivas.

**Conflicts of Interest:** The authors report no conflicts of interest.

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
