# Peer review of "Is Agricultural Emissions Mitigation on the Menu for Tea Drinkers?"

_sustainability, doi:10.3390/su11184883_

Round 1

Reviewer 1 Report

Review sustainability-551920

Reviewer/Report #1

GENERAL COMMENTS

This manuscript presents a study that assessed the willingness to pay of the consumers regarding the agricultural GHG mitigation using green tea as a study case. The study is scientifically sound, the survey and its analysis were conducted appropriately and statistical analyses are described with transparency and details. The subject suits perfectly in the scope of Sustainability.

For all those reasons, I recommend the acceptance of this manuscript for publication in Sustainability after the major revisions specified below (see SPECIFIC COMMENTS) will be applied.

SPECIFIC COMMENTS

Comment (C) 1

Technical but important when submitting a manuscript:

Please, input line numbers on next manuscript submissions to journals, it is the baseline procedure and it facilitates the task for the reviewers to input their comments.

C2

LINE:

Participants were WTP for agricultural GHG mitigation, but there was substantial variability by income, gender, and geographic location.

Please reformulate as it would be read as "Participants were willingness to pay for agricultural GHG mitigation" and it is not clear.

One of the main finding in the discussion is “Specifically, males, participants from the Midwestern study site and lower-income individuals had higher WTP for agricultural GHG mitigation in tea production.” and “In the latent class analysis, higher household income was significantly associated with the probability of members to latent class two, the class whose members were sensitive to the price levels presented in the choice experiment but no other product attributes.”…and with a low WTP for GHG mitigation

These two sentences represent important findings in this study and must appear in the abstract as it describes better the findings than only mentioning “but there was substantial variability by income, gender, and geographic location”. Being more specific on describing the main findings in abstract is more appealing for the readers. Please solve this issue.

C3

LINE:

In 2010, approximately 7,000 products in the U.S. exhibited an environmental claim, including 89 products asserting to be carbon neutral [4]. These labels are increasingly found on foods and beverages. Such items can be labeled as organically or locally grown, genetically modified organism-free, fairly traded, etc. Numerous studies have shown that certain consumers are willing to pay a premium for such labels and production attributes (Aoki and Akai, 2012; Breusted, 2014; Grunert et al., 2014; Shewmake et al., 2015; Peschel et al., 2016).

It is not clear here if there is actual knowledge on the WTP for climate-friendly products. It is written that climate labels (carbon neutral) already exist for some products and that clients are willing to pay more for these. Please make the gap in knowledge clearer than it is described here already in the beginning of the introduction. Describe firmly the gap in the knowledge. For now, one could think that there is no need for the study presented in the manuscript.

C4

LINE:

increasing consumer demand for climate friendly foods could help mitigate future climate impacts and shocks to agriculture

Please correct grammar: it would be read as "increasing consumer demand for climate friendly foods could ... shocks to agriculture.

C5

LINE:

The objective of the present study was to determine if U.S. specialty food and beverage shoppers are …

Please be more specific in the objectives on the type of food and beverage targeted by the study. The reason is that the readers need to understand that this study is unique and brings something new. The introduction mentions studies performed in Europe while only two studies were done in the U.S. on this topic for apple and beef. What is unique about your study that is covered by other studies. This should be stated clearly.

The case study is on green tea, which is clearly stated in the title. Then the objectives must also indicates that the study is on green tea. In the present state, the objectives are written in a way that the study cover several types of food and beverage while only green tea is targeted. The objectives must be more specific.

C6

LINE:

Consumers were required to taste both spring and monsoon harvested samples because a concurrent study was assessing the sensory characteristics and consumer WTP for tea harvested in different seasons.

Please cite this concurrent study if it is published.

C7

LINE:

After tasting tea samples, participants read an instruction sheet prior to beginning the choice experiment. The instruction sheet explained that participants should consider a real-world setting where they would be buying a box of one of the sampled teas containing 18 sachets and to choose the product they preferred most in this setting, or they could choose the “I prefer neither option”.

Is this related to Fig.1? Please indicate if it is so.

C8

LINE:

The phenomena included: “pollution/emissions from business/industry”; “people driving cars”; “use of coal/oil by utilities/electric companies”; “use of aerosol spray cans”; “chemicals that destroy the ozone layer”; “nuclear power generation; people heating/cooling their homes”; “destruction of forests”. Up to two points were awarded per question when participants answered correctly. Participants could obtain a score of zero if they answered none of the questions correctly, or up to 18 if they answered all questions correctly.

This is unclear. A total of seven questions is indicated here in "…". If a max of 2 points is given to a correct answer, then the max score is 14, not 18. Please correct this issue.

C9

LINE:

Data analysis was conducted using Stata 15.1/SE.

Please indicate this important detail at the beginning of the section on Data analysis.

C10

LINE:

68.0% of the sample had earned a college or degree or higher. 24.0% of the sample was classified as high income.

Please do not start a sentence with a number. Please correct.

C11

LINE:

The mean knowledge score was 11.6 (SE 0.022) and the mean score on risk perception measures was 4.5 (SE 0.011).

Knowledge score is not accurately defined in the methods. Please be more specific on the definition. What is the meaning of the scores? How 11.6 and 4.5 behave on a scale if these scores can be compared on a scale.

C12

Table 2

“Some college”

Please be more specific on defining this label.

The term race can be delicate to use and could affect some people. Please just use Ethnic group for instance.

Why hispanic/latino group is not within the ethic group?

C13

LINE:

The coefficient for price was also negative, as expected, indicating that an increase in the price of the tea decreases the probability of selecting a tea alterative

"as expected". Where are the hypotheses in order to write this? Please solve this issue.

“alterative”: correct for alternative

C14

Paragraph starting with "Table 3 includes results from Model 2, the… "

Please spit this paragraph in two: one for the scores from Model 2 and the other for the scores of Model 3.

C15

LINE:

(β=−0.236,p<0.05)

No need to repeat the values when they are already in the table 3. Please correct this issue for all duplicates table-text found in this paragraph and through all the text.

C16

Table 3

What do you specifically define as the "results" in the caption? The results are the numbers presented in the table. What is the real and appropriate name for those numbers? Scores (+/- SE)? Please be specific.

Why in Model 2 all demographic characteristics are in the same group? For instance, how is it possible to compare age class with education level with scores being negative and positive? Please clarify this issue. The problem may be only on how the lines are separated in the table. In the actual state, it seems that all demographic characteristics are compared in the same group. Please solve this issue.

Observations

Why it is 9,120? The total amount of participants was 380? Please explain or be more specific.

C17

Table 4.

Same question than for table 3. Why n = 9,120? There were 380 participants? Please explain or be more specific.

C18

LINE:

The optimal number of latent classes was determined to be six, based on the AIC, CAIC, BIC and Log-likelihood values for latent class models with two to six classes. See table S1 for AIC, CAIC, BIC, and log-likelihood values for classes two through six.

The likelyhood values resulted in the choice of six classes. Concretely what differentiates and defines these classes? What are the characteristics of these classes? Please be more specific.

C19

Table 5

The association is presented as coefficients. Please be specific on this point in the caption. Define what is meant by associations as it is presented with the coefficients.

C20

Paragraph starting with "Table 6 shows the association ..."

Tables 6 and 7 and the related paragraph must be presented before Table 5 as it seems to define the classes. Moreover, based on Tables 6 and 7, please be more specific on defining what each class represents, this is related to C18. For instance, it is mentioned that class 2 is better defined with people with high incomes. Well done but what defines the other classes. This is crucial to better understand the relationships done in Table 5 and the discussion section. Please restructure the section (Tables 6 and 7 and text before Table 5) and define the classes before introducing the results in Table 5.

Author Response

Please see the attached document for responses to your comments and responses to Reviewer 2 comments.

Reviewer 2 Report

This is an interesting study that makes a useful contribution to understanding consumer motivation for low carbon footprint foods. I find the manuscript acceptable for publication with a few minor changes

Abstract line 38 makes the claim that no prior study has examined the interaction between climate knowledge and willingness to purchase low carbon footprint foods. I don't believe this is valid, see for example Echeverria et al 2014 British Food Journal 116(2):186-196; Zhong and Chen 2019 MDPI Sustainability 11:592; Kim et al 2016 IFAMA 19(4) DOI: 10.22434/IFAMR2015.0095; Feucht and Zander Int J Food System Dynamics 2017 DOI: http://dx.doi.org/10.18461/pfsd.2017.1738

Abstract line 44, WTP (i.e. willingness to pay) does not fit the sentence; substituon would read, " Participants were willingness to pay for..."

Keywords. Maybe add "Carbon footprint". There is also a bit of duplication (climate change, climate change knowledge, climate change risks) that could be simplified

I think the authors could interact with relevant literature more completely. In addition to references above, consider: Elofsson et al 2016 Food Policy 58:14-23; Michaud et al 2013 Eur Rev Agr Econ 40(2):313-329; Lim-Camacho et al Regional Environmental Change 2017, Volume 17, Issue 1pp 93–10

Author Response

Please see the attached document for responses to your comments and responses to Reviewer 1 comments. 

Round 2

Reviewer 1 Report

The authors responded to all the comments I had on the original submission. Therefore, I recommend the acceptance of this manuscript for publication in Sustainability.